# Optimisation on the Performance of Bubble-Bursting Atomisation for Minimum Quantity Lubrication with Vegetable Oil Using Computational Fluid Dynamics Simulation

**DOI:** 10.3390/ma15124355

**Published:** 2022-06-20

**Authors:** Pin Han Yap, Jaharah A. Ghani, Wan Mohd. Faizal Wan Mahmood

**Affiliations:** Department of Mechanical and Manufacturing Engineering, Faculty of Engineering and Built Environment, Universiti Kebangsaan Malaysia, Bangi 43600, Selangor, Malaysia; p110223@siswa.ukm.edu.my (P.H.Y.); faizal.mahmood@ukm.edu.my (W.M.F.W.M.)

**Keywords:** computational fluid dynamic, bubble-bursting atomisation, minimum quantity lubrication

## Abstract

Sustainable and green machining technologies have become a welcomed topic in the manufacturing industries. One of the emerging sustainable technologies is minimum quantity lubrication (MQL). In this study, the optimisation and study of the bubble-bursting atomisation system applied to MQL machining is carried out through the computational fluid dynamics (CFD) simulation approach. Vegetable oil is selected as the cooling lubricant in this study. The performance of the bubble-bursting atomisation system is improved by alternating air inlet velocity and the gap distance between the inlets of bubble production. A velocity of 0.1 ms^−1^ is suitable for the air at the inlets for the bubble production, whereas 10 ms^−1^ is suitable for the velocity of the air at the inlet, where the droplets of vegetable oil are guided to the nozzle. Besides that, a 50 mm gap distance between the air inlets for the production of bubbles is able to avoid the occurrence of bubble coalescence. Under these conditions, optimal bubble sizes of 2–3 mm can be achieved, with a higher probability of nano-sized droplets being present in these ranges. Furthermore, a higher rate and smaller size of vegetable oil droplets escaping the atomisation chamber and reaching the machining zone will be generated. Thus, the performance of the MQL machining can be improved.

## 1. Introduction

Machining is a significant process in various manufacturing industries such as automotive and aerospace. According to ReportLinker, the global machine tool market size will continue to grow at a compound annual growth rate (CAGR) of 4% through 2027. This means that demand from end-use industries such as the automotive and electrical industries is still increasing, indicating that global machining still has a growing trend [1]. Conventionally, flood cooling or wet machining has been used to improve the machinability of metal cutting. However, this method causes a large amount of cutting fluid to be wasted and increases environmental pollution [2]. Not only that, many companies are now being forced to implement strategies in order to reduce the amount of cooling lubricants because of the increase in costs for waste disposal [3]. Minimum quantity lubrication (MQL) is one of the technologies that is able to minimise pollution and waste disposal compared to the conventional flood cooling system. MQL utilises compressed air to atomise a very small amount of cutting fluid (approximately 30–100 mL/h,) and the mixture is then targeted to the workpiece for effective lubrication. It is believed that MQL is more sustainable and environmentally friendly compared to conventional methods, which have brought disadvantages economically, environmentally, and even to human health. The global manufacturing industry has faced serious threats with respect to sustainable development. MQL is one of the proposed green cutting lubrication technologies in the context of sustainable manufacturing [4]. MQL technology has been proven to improve the performance of machining [5,6]. There are also many articles that have shown by their results that the advancement of MQL outperforms the independent MQL system by generating better surface quality and reducing tool wear and cutting force [7,8,9,10,11,12,13,14,15,16,17,18,19]. Besides that, there have been a number of studies that have shown that MQL is indeed able to increase machining performance, especially when nanofluids are involved. Sayuti et al. reported that MQL systems using nanofluids have better results compared to conventional flood cooling systems [20]. Dambatta et al. also reported that SiO2 nanoparticles in vegetable oil possess good lubrication effects in the MQL-grinding process [21]. Dry cutting also belongs to sustainable manufacturing, but machining at a high cutting speed in a dry condition, in particular for difficult to-cut materials such as titanium alloy and hardened steels, causes rapid tool wear, high forces and cutting temperature, and unfavourable chip formation [22,23,24].

The unique properties of nanoparticles have brought many benefits to society. Although nanofluids are highly effective in machining, they have also been criticized by others when it comes to issues such as human health, environmental pollution, storage stability, waste disposal, biodegradability, oxidativeness, and cost. Nanoparticles have a larger surface area and particle number per unit mass. These properties increase the toxic potential as the potential for reactivity is also increased [25]. The areas of concern are workplaces such as laboratories, production facilities, or operational facilities in which engineered nanomaterials are involved. Sometimes, the nanoparticles are not properly recycled or disposed of after machining processes. They are released either through air or water during production or as a waste by-product, which in the end will be deposited into soil, water, or living organisms. Aerosols generated from a bioreactor may contain harmful biological substances, which can potentially lead to environmental hazards [26]. Not only from the environmental aspect, human health issues also usually occur in laboratories involving nanotechnology research or manufacturing sites involving nanoparticles. Possible health impacts from exposure to nanoparticles are the development of fibrosis and other pulmonary effects after short-term exposure to carbon nanotubes. Additionally, the activation of platelets, the enhancement of vascular thrombosis, and the translocation of nanoparticles to the brain via the olfactory nerve may also occur [27].

The manufacturing industry is always looking for better solutions to overcome these machining problems posed by nanofluids. Nanoaerosol (NAs) lubricants may be the solution to these problems, as NAs have the potential to improve machining performance by penetrating the tool–chip interface, helping to create lubrication effects that are largely similar to nanofluids. This could serve as a starting point for investigating the effectiveness of NA lubricants in MQL machining, if the lubricant used can be directly converted into NAs when using a compatible liquid atomization system and applied to MQL machining rather than using nanofluid.

In recent years, some researchers have studied the droplets generated after the bubble burst under different experimental conditions [28,29,30]. From their research, the bubble-bursting atomisation method possesses very high potential in generating NAs and applicability to MQL machining. Bubble bursting is common in many fields, including nuclear engineering, the chemical industry, and marine aerosol environments. Bursting bubbles will basically produce two types of droplets, which are film drops and jet drops. Film drops are produced from the rupture of the bubble film, whereas jet drops are produced from the breakup of the vertically rising water jet resulting from the collapse of the bubble cavity [31]. The sizes of jet drops are in the order of one-tenth the bubble diameter [32]. Bubble plume’s life stages can be separated into four processes, which are (1) formation, (2) injection, (3) rise, and (4) senescence [33]. Bubble bursting plays a key role in the mass flux between the sea surface and the atmosphere as a source of aerosol droplets in the air [34]. Aerosols are dispersed into the atmosphere via disintegration of the bubble film and the fragmentation of the upwardly directed liquid jet formed when the bubble film collapses. The failure of surface tension to maintain the centripetal forces can lead to the generation of film drops [35].

In this work, a series of simulations of bubble-bursting atomisation is conducted in order to optimise the quality of film drops before the film drops are delivered to the workpiece-tool machining region. The performance of bubble-bursting atomisation for MQL by using vegetable oil as the fluid is studied and optimised by using computational fluid dynamics (CFD) simulation. Different factors that influence the performance of bubble-bursting atomisation such as air inlet velocity and the gap distance between the air inlets are studied and analysed.

## 2. Materials and Methods

Figure 1 below shows the flow diagram of the bubble-bursting MQL system. The focus of this work is the atomisation chamber only. The discussion of the nozzle’s role is excluded in this study. This is because the nozzle is located at the outlet of the atomisation chamber. The formation of the nanoaerosols inside the atomisation chamber are more important in this article. The visualisation of how the bubbles are formed and the size of the bubbles is the primary objective. The nozzle part will only be included when it comes to machining, where the workpiece-tool region is involved.

Prior to the simulation study, the fluent module was applied for the workbench. A 2D model was created by using SpaceClaim. Figure 2 shows the flow chart of the simulation study. The steps involved were creating a fluent module for the workbench, importing and defining the model surface, meshing, setting solver parameters, as well as material parameters, running calculations, and analysing the results.

After the 2D model of atomisation chamber is created by using SpaceClaim, the surface model is then meshed, as shown in Figure 3.

The important meshing properties are summarised in Table 1 below.

An element size of 1.2 × 10^−3^ m was chosen because it provides an optimum result in terms of stability and animation results after several trials in manipulating the element size. The generation of air bubbles at the inlets and their interaction between the wall and outlets can be observed at a higher resolution and produce a better result when edge sizing is applied. For face meshing, the triangle method was chosen instead of quadrilateral because only the 2D model is involved and the shape is not complicated; thus, only low computational power is required. Overall, the simulation was stable and converged well by applying these meshing properties.

The naming of the boundary conditions is given in Figure 4 below.

After meshing and the specification of the boundary conditions, all the required parameters were then set up by using FLUENT. All the information in FLUENT is shown in Table 2, Table 3, Table 4, Table 5 and Table 6.

In Table 2, the pressure-based type was chosen because no high-speed compressible flow was involved. A transient time was selected because the loads in this simulation were a function of time. Gravity must be considered in this simulation work. VOF was used because two immersible fluids were involved. An explicit formulation is easier to program, and a convergence result can be obtained if the step size is carefully adjusted. The number of Eulerian phases was 2, as only 2 fluids were involved in this simulation, which is vegetable oil and air. The realizable k-epsilon model provides improved prediction for flows involving rotation, boundary layers under strong adverse pressure gradients, separation, and recirculation. Enhanced wall treatment was used because it blends the separate models in the two-layer approach by use of a damping function so that the transition between the two is smoother.

**Table 3 materials-15-04355-t003:** Materials involved.

Type of Fluids	Density (kg/m^3^)	Viscosity (kg/(ms))
air	1.225	1.7894 × 10^−5^
vegetable oil	910	0.035

Surface tension coefficient = 0.032 N/m.

Table 3 shows the values of the density and viscosity for air and vegetable oil.

**Table 4 materials-15-04355-t004:** Boundary conditions.

Zone	Properties
outlet	pressure outlet
wall	stationary wall, no slip

In Table 4, the pressure outlet is recommended by ANSYS since the velocity of the inlet is known. A stationary wall was chosen because it is a fixed wall. A no-slip wall was chosen because an assumption was made that the speed of the fluid layer in direct contact with the boundary is identical to the velocity of this boundary.

**Table 5 materials-15-04355-t005:** Solution methods.

Scheme	SIMPLE
Gradient	Least-squares-cell-based
Pressure	PRESTO!
Momentum	Second-order upwind
Volume fraction	Geo-reconstruct
Turbulent kinetic energy	First-order upwind
Turbulent dissipation rate	First-order upwind
Transient formulation	First-order implicit

In Table 5, the SIMPLE algorithm is more favoured for convergence compared to SIMPLEC in the case of high mesh skewness. The least-squares-cell-based scheme is more accurate than the Green–Gauss-based one. PRESTO! is generally much better unless there is stability issue. Geometric reconstruction is a scheme that represents the interface between fluids using a piecewise-linear approach. In ANSYS, this scheme is the most accurate and is applicable to general unstructured meshes. Second-order upwind momentum is much more accurate than first-order, but has stability issues. However, a first-order scheme can have better absolute accuracy than a second-order scheme.

**Table 6 materials-15-04355-t006:** Boundary conditions of each simulation.

Simulation No.	Air Inlet Velocity (ms^−1^)	Air Inlet_2 Velocity (ms^−1^)	Distance of Gap between Air Inlets (mm)
1	1.0	1.0	15
2	0.1	0.1	15
3	0.1	1.0	15
4	0.1	10.0	15
5	0.1	10.0	50

There were in total five simulations carried out in this work, and their respective boundary conditions are summarised in Table 6. The results of the simulation converged well by using all the parameters shown in Table 1, Table 2, Table 3, Table 4, Table 5 and Table 6.

## 3. Results and Discussion

After all of the required parameters and conditions were specified, Simulation 1 was then run, and the air-phase contour is shown in Figure 5. The scale of the air-phase contour is in volume fraction, where values of 1 (red) and 0 (blue) represent air and oil, respectively.

The objective of the bubble-bursting atomisation system is to produce fine oil droplets with a uniform size consistently and continuously. However, from Figure 5, the production of air bubbles is not consistent, and a large amount of vegetable oil escaped through the outlet of the atomization chamber after only one second. This is definitely not the case expected. The air inlet velocity of 1.0 ms^−1^ is believed to be too high for the generation of the bubbles with uniform sizes. Based on Simulation 1, the air inlet velocity of 1.0 ms^−1^ is not suitable in producing uniformly sized bubbles and a lower inlet air velocity should be used.

After Simulation 1, Simulation 2 was conducted. All the parameters and boundary conditions remained the same. The only change in Simulation 2 was the velocity of the air inlet and air inlet_2, which changed from 1.0 ms^−1^ to 0.1 ms^−1^. Simulation 2 was then executed. The results from Simulation 2 are shown in Figure 6 below.

From Figure 6, it can be seen that, this time, the vegetable oil did not spread all over inside the atomisation chamber, and the generation of bubbles became more consistent as compared to Simulation 1. From the view of the bubble generation, the bubble formation is more obvious and clearer. However, a phenomenon happened where the bubble produced started to converge at the middle after 0.5 s. The occurrence of the convergence is most probably due to the difference in velocity, where the zone of high velocity (tip of the air inlet) creates a lower pressure zone at the middle (refer Figure 7). This explanation is based on Bernoulli’s Principle.

In other words, this type of convergence is known as bubble coalescence. Bubble coalescence is a process where two or more adjacent bubbles combine together and form a larger bubble. The majority of the bubble formation in Figure 6 is larger than 5 mm in diameter. A bubble of a size larger than 5 mm will reduce the efficiency of the atomisation system for MQL application. Based on Cipriano et al., the optimal bubble size is around 2–3 mm for greater potential efficiency of the film drop mechanism. The estimated production of the film drop reaches a peak maximum for bubbles of roughly 2 mm in diameter [36]. The film drop is the main source of the nanoparticles. Bubbles smaller than 1 mm or larger than 5 mm are relatively inefficient for the atomisation system. From Figure 7, it was noticed that there was a recirculation flow just above the surface level of the vegetable oil (Zone A). The formation of the recirculation zone is caused by the existence of an axial-symmetric vortex structure due to the sucking effect of the negative pressure zone, which leads to the reverse flow of the fluid [37]. Recirculation flow increases the chances for the attraction between the particles due to the attractive capillary force. Thus, particles will coalesce, and their size will increase [38]. If the size of the particles above the surface of the vegetable oil exceeds a certain weight, those heavy particles will eventually drop back into the vegetable oil. As a result, the production of nanoparticles will be affected and reduced.

Besides that, particle tracking was also conducted in order to study the pathway flow of the particles and ensure the existence of droplet particles at the outlet of the atomization chamber. There were 77,000 particles of vegetable oil (density = 910 kgm^−3^) of a mean size of 5 µm injected from the air inlet by using the Rosin–Rammler diameter distribution method. Another purpose for injecting these particles is to reflect the real-life scenario, where these particles can be represented by film drops produced after the bubble bursts just above the vegetable oil level. The results are shown in Figure 8 and Figure 9 below.

The purpose of showing Figure 8 is to confirm that there were actually injected particles escaping from the outlet of the atomisation chamber successfully. Figure 9 shows the pathway flow of the injected particles.

It can be seen that, although there were particles detected at the outlet of the atomisation chamber, the particles slowed down halfway and dropped back to the surface of the vegetable oil due to gravity instead of flowing straight to the outlet. This provides crucial information that there is insufficient air velocity at air inlet_2. Simulation 3 was then set up, and the results of Simulation 3 are shown in Figure 10 and Figure 11 below.

After the velocity of air inlet_2 was increased from 0.1 ms^−1^ to 1.0 ms^−1^, it can be seen that the effect of recirculation flow just above the surface level of the vegetable oil and the effect of gravity on the injected particles reduced significantly, as shown in Figure 10 and Figure 11, respectively. As a result, the concentration and the chances of smaller particles exiting the outlet of the atomisation chamber can be greatly increased.

Besides the escaping of particles at the outlet, in machining, the velocity of particles plays an important role in lubrication. The particles leaving the outlet must gain enough kinetic energy in order to penetrate into the working zone of the machining for effective lubrication. Furthermore, based on Nath et al., a higher droplet velocity allows gradual mixing or diffusion of the droplets into the gas jet. This condition helps in the early development of the far-field region and leads to the improvement of the tool life for machining [39]. Next, a higher velocity of air inlet_2 was applied in Simulation 4, and the results are shown in Figure 12 and Figure 13 below.

It can be noticed that the high air velocity zone shown in Figure 12 actually created a low-pressure zone, where the film drops produced just after the bursting of the bubble will act against gravity and be directed toward the outlet of the atomisation chamber with the help of the high-velocity air from inlet_2.

Besides that, from Figure 13, it can be seen that up to 67% of the injected particles in Simulation 4 were detected at the outlet of the atomisation chamber in only 1 s, whereas in Simulation 3, only up to 34% of the injected particles arrived at the outlet of the atomisation chamber in 2.75 s. This means that the rate of particles escaping the atomisation chamber and reaching the machining zone will be higher in the long run for the conditions in Simulation 4 as compared to Simulation 3. A value of 10 ms^−1^ is suitable for the velocity of the air at inlet_2. With this value, the recirculation zone is reduced significantly and a higher droplet velocity is obtained. These conditions help in the early development of the far-field region, and the production of the nanoparticles can be increased. As a result, we have a higher chance of improving the tool life for machining.

There is one more problem left to be solved, which is the convergence of the air bubbles (bubble coalescence) in the vegetable oil, as shown in Figure 6. It was then decided to increase the distance of the gap between the air inlets and see whether the problem could be solved. Simulation 5 was then conducted, and the result of Simulation 5 is shown in Figure 14 below.

From Figure 14, the bubbles generated did no longer converged, which means that the rate of bubble coalescence reduced significantly. As a result, the size of the bubble produced will be more uniform and maintained within 2–3 mm. With the conditions in Simulation 5, the performance of the machining can be improved with respect to the air inlet velocity and gap distance between the air inlets.

## 4. Conclusions

The bubble-bursting atomisation system applied in MQL was optimised and studied through CFD simulation with respect to various parameters including the air inlet velocity and gap distance between the air inlets. A vegetable-oil-based lubricant was selected as the fluid used because it is harmless and biodegradable and has excellent lubricity. After five simulations with different input parameters, it can be concluded that the velocity of the air at the inlets for bubble production cannot be too high. If not, the vegetable oil will spread all over the atomisation chamber randomly and will be highly unstable. A velocity of 0.1 ms^−1^ is suitable for the air at the inlets for bubble production. A velocity of 10 ms^−1^ is suitable for the air at the inlet, where the droplets of vegetable oil are guided to the nozzle. This condition will increase the rate of droplets escaping the atomisation chamber and reaching the machining zone in the long run. The size of the bubbles can be maintained within the optimal diameter range, which is 2–3 mm, by increasing the gap distance between the air inlets for bubble production to 50 mm. A bubble diameter size of around 2–3 mm possesses greater potential efficiency for the film drop mechanism. However, there are unforeseen potential errors, which are able to affect the results of this simulation when compared to real-life scenarios. The liquid and surrounding temperatures might play a role in affecting the NA lubricants produced. The physical properties of the vegetable oil used will also slightly influence the results. Furthermore, air leaking in the real-life scenario might also affect the pressure inside the atomisation chamber. Thus, the results can also be affected. This article is purely based on a simulation, and it has not yet been validated experimentally because of the lack of suitable instruments for measuring the droplets’ size at the nano-scale. More research must be conducted on the bubble-bursting atomisation method in order to improve the quality of nanoaerosol lubricants. Nanoaerosol lubricants will become the future trend and make significant contributions to the sustainability of the global machining industry in the future.

## Figures and Tables

**Figure 1 materials-15-04355-f001:**
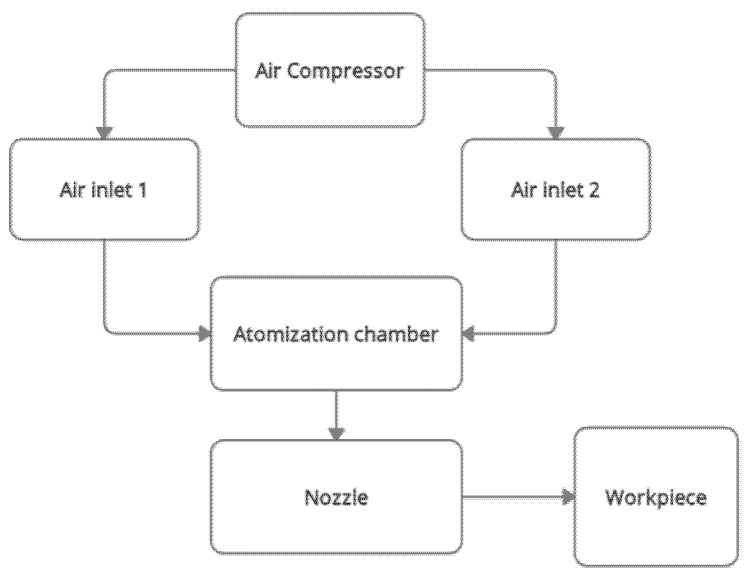
Flow diagram of the bubble-bursting MQL system.

**Figure 2 materials-15-04355-f002:**
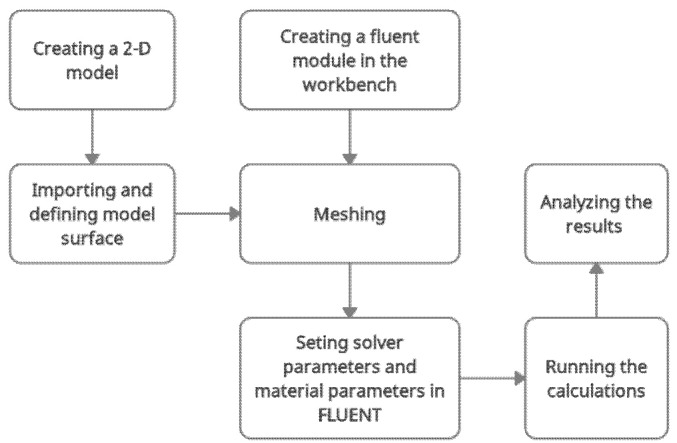
Simulation flow chart.

**Figure 3 materials-15-04355-f003:**
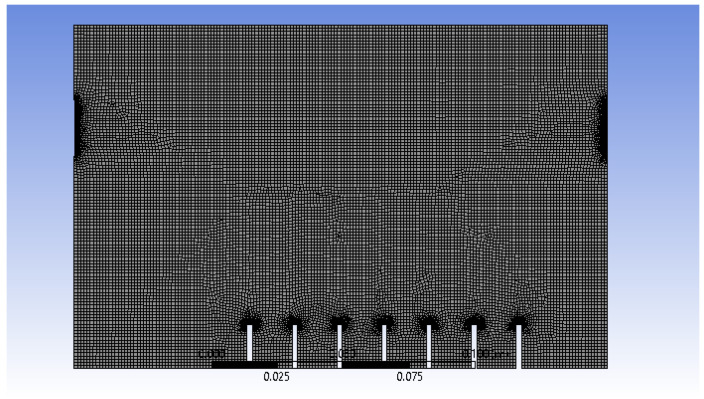
Meshing of 2D model of atomisation chamber.

**Figure 4 materials-15-04355-f004:**
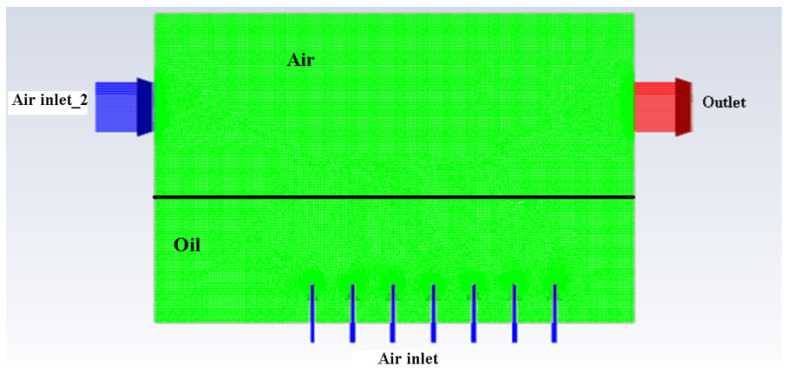
Naming of boundary conditions of atomisation chamber.

**Figure 5 materials-15-04355-f005:**
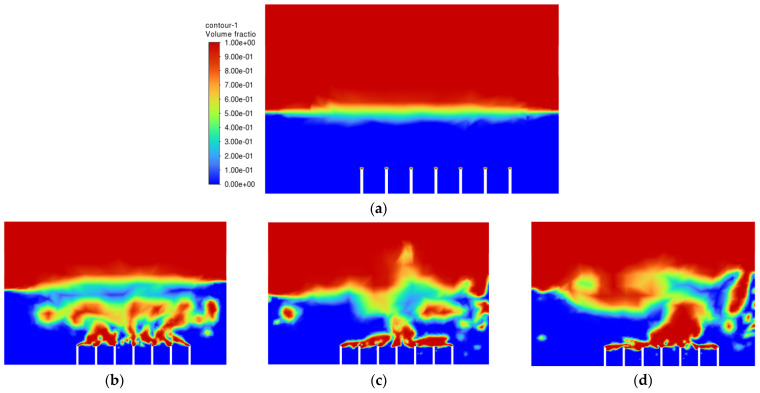
Air-phase contour of Simulation 1: (**a**) 0.00 s; (**b**) 0.25 s; (**c**) 0.50 s; (**d**) 0.75 s; (**e**) 1.00 s; (**f**) 1.25 s; (**g**) 1.50 s; (**h**) 1.75 s; (**i**) 2.00 s; (**j**) 2.25 s.

**Figure 6 materials-15-04355-f006:**
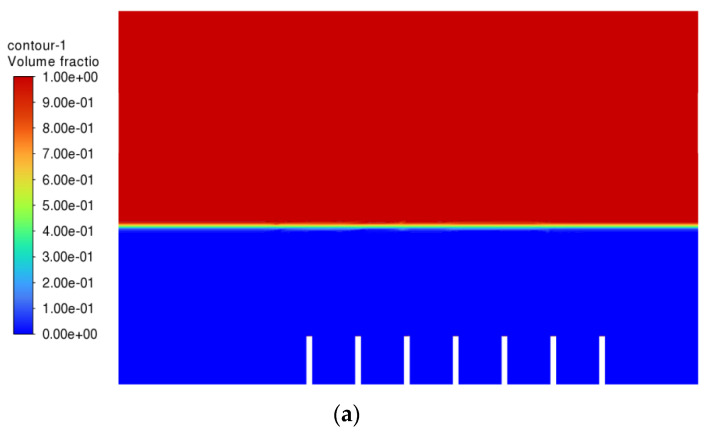
Air-phase contour of Simulation 2: (**a**) 0.0 s; (**b**) 0.5 s; (**c**) 1.0 s; (**d**) 1.5 s; (**e**) 2.0 s; (**f**) 2.5 s; (**g**) 3.0 s; (**h**) 3.5 s; (**i**) 4.0 s; (**j**) 4.5 s.

**Figure 7 materials-15-04355-f007:**
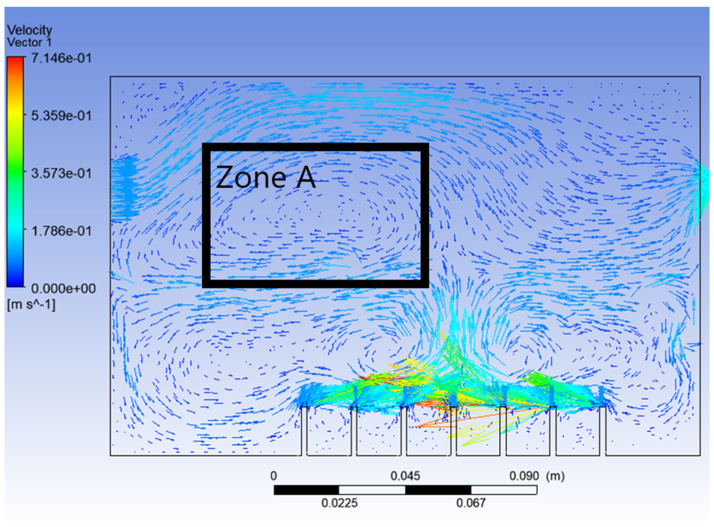
Velocity vector of Simulation 2.

**Figure 8 materials-15-04355-f008:**
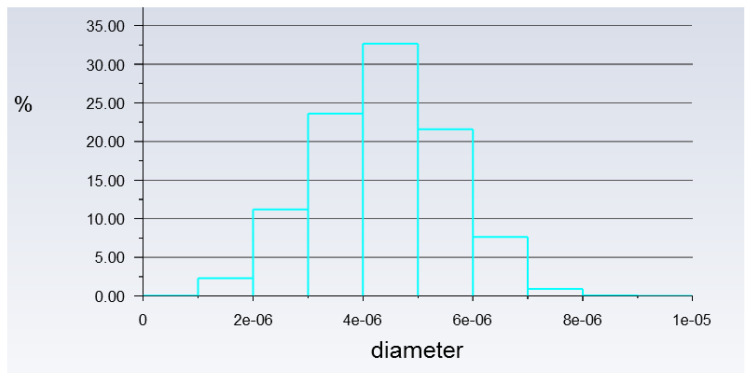
Histogram of particles at the outlet of Simulation 2. *Units of diameter in m.

**Figure 9 materials-15-04355-f009:**
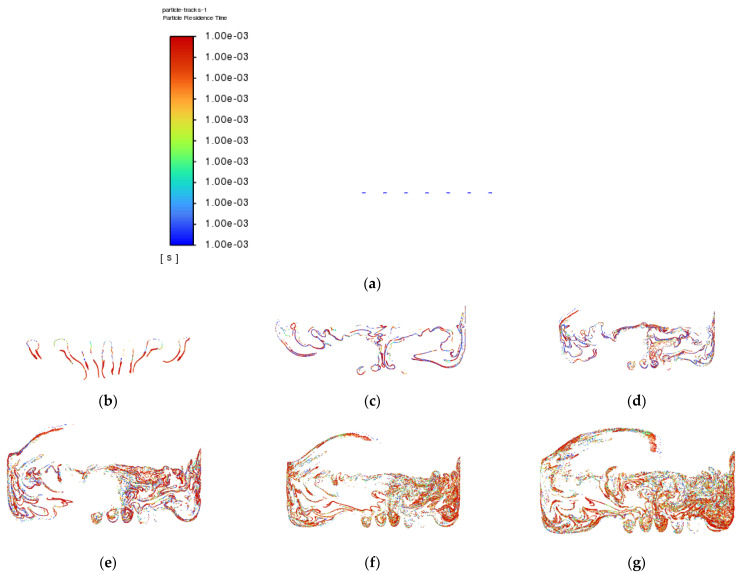
Particle tracking of Simulation 2: (**a**) 0.0 s; (**b**) 0.5 s; (**c**) 1.0 s; (**d**) 1.5 s; (**e**) 2.0 s; (**f**) 2.5 s; (**g**) 3.0 s; (**h**) 3.5 s; (**i**) 4.0 s; (**j**) 4.5 s.

**Figure 10 materials-15-04355-f010:**
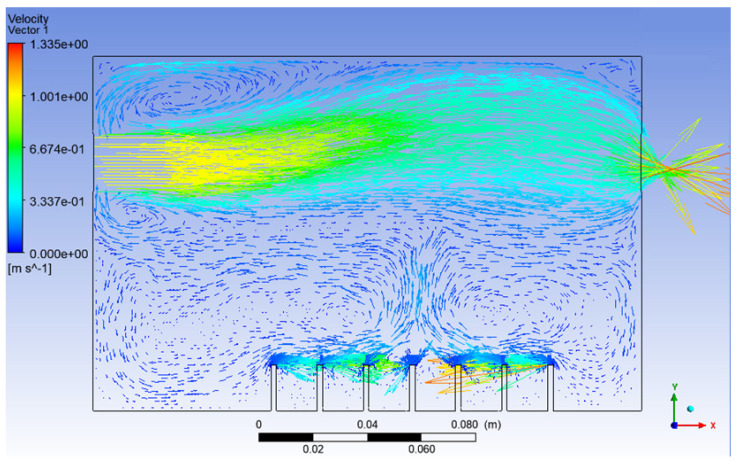
Velocity vector of Simulation 3.

**Figure 11 materials-15-04355-f011:**
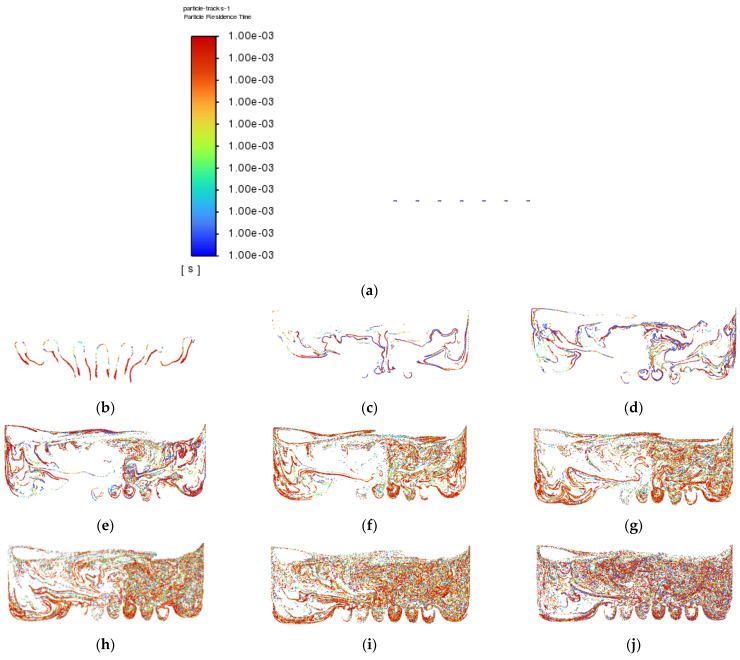
Particle tracking of Simulation 3: (**a**) 0.0 s; (**b**) 0.5 s; (**c**) 1.0 s; (**d**) 1.5 s; (**e**) 2.0 s; (**f**) 2.5 s; (**g**) 3.0 s; (**h**) 3.5 s; (**i**) 4.0 s; (**j**) 4.5 s.

**Figure 12 materials-15-04355-f012:**
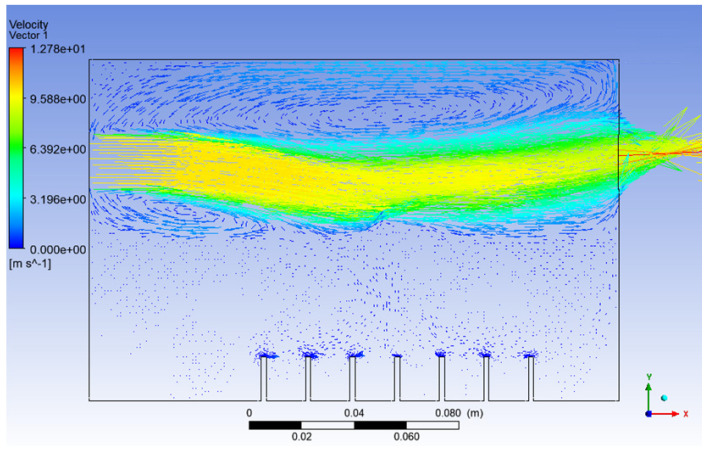
Velocity vector of Simulation 4.

**Figure 13 materials-15-04355-f013:**
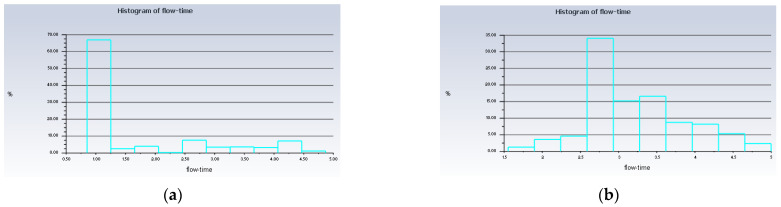
Histogram of the low time of the injected particles. (**a**) Simulation 4; (**b**) Simulation 3.

**Figure 14 materials-15-04355-f014:**
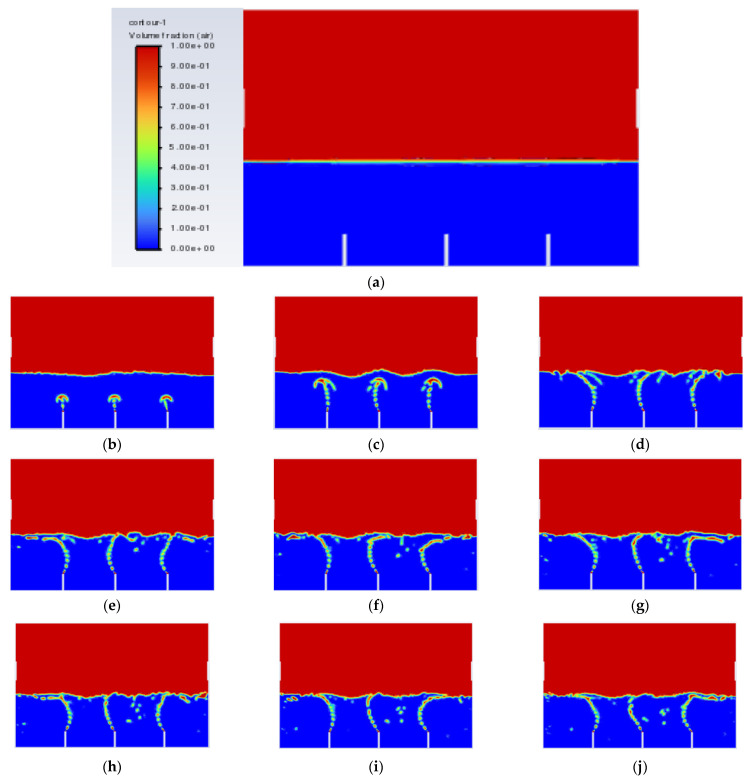
Air-phase contour of Simulation 5: (**a**) 0.00 s; (**b**) 0.25 s; (**c**) 0.50; (**d**) 0.75 s; (**e**) 1.00 s; (**f**) 1.25 s; (**g**) 1.50 s; (**h**) 1.75 s; (**i**) 2.00 s; (**j**) 2.25 s.

**Table 1 materials-15-04355-t001:** Meshing properties of 2D model of atomisation chamber.

Element Size	1.2 × 10^−3^ m
Edge Sizing 1	Edges selected: air inlets and outlets
	Number of divisions: 100
Edge Sizing 2	Edges selected: Wall
	Number of divisions: 50
Face meshing	Triangle: best split

**Table 2 materials-15-04355-t002:** General models.

Type	Pressure-based
Time	Transient
Gravity	−9.81 ms^−1^ (y-direction)
Model	Volume of fluid (VOF)
Formulation	Explicit
Body force formulation	Implicit body force
Number of Eulerian phases	2
Viscous	k-epsilon (2 eqn)
k-epsilon model	Realizable
Near-wall treatment	Enhanced wall treatment

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
