# Peer review of "Optimisation on the Performance of Bubble-Bursting Atomisation for Minimum Quantity Lubrication with Vegetable Oil Using Computational Fluid Dynamics Simulation"

_materials, 2022, doi:10.3390/ma15124355_

Round 1

Reviewer 1 Report

In this study, optimisation and study of the bubble-bursting atomisation system applied on MQL machining is carried out through computational fluid dynamic (CFD) simulation approach. Here are the suggestions before it could be accepted.

1. The literature should be updated, more literature should be in recent three years.

2. In the introduction, the disadvantages of the references should be summarized clearly to emphasize the importance of this work.

3. In Figure 8. The unit of diameter is “m”, may it is better to change it into “μm” or “mm”.

4. The resolution of Fig.5 and Fig.6 should be improved.

5. How to validate the results? It should be compared with experimental results.

6. Is the format of conclusion right? Please accord to the requirements of the journal.

Author Response

The respond to reviewer comments is in the attached file

Reviewer 2 Report

In its current form, the manuscript has some shortcomings that need to be addressed. Comments to improve the manuscript are as follows:

1. The first major drawback of this research is that potential errors have not been analysed and discussed.

2. Another major shortcoming of this research is that no experimental research has been conducted to support the simulation results.

3. At the end of the Introduction section, highlight the scientific contribution of your research. Materials is a top journal, it is not enough to state the following: "In the author’s cognition, application of bubble-bursting atomisation technology on MQL system with vegetable oil has not yet been studied". The authors did not study, but only simulated.

4. Section 2. Materials and Methods must be significantly corrected and supplemented. The choice must be explained in detail. Time saving cannot be a scientific explanation for the selection of input parameters.

5. Why the analysis and discussion of the nozzle’s role is excluded from the study when it is very important.

6. The parameters shown in Tables 2-6 must be explained in detail. Each choice must be explained in detail in order for the applied methodology to be universal.

7. How the authors of previous research [6-18] can use your results. Elaborate in detail the possibilities of practical application of your methodology.

8. The conclusions should certainly state the limitations of the methodology and directions of future research.

Author Response

Respond to the reviewer comments is in the attached file

Round 2

Reviewer 1 Report

It can be published in this form.

Reviewer 2 Report

The article has been corrected and updated. I suggest accepting the article in its current form.